# Multiple mini-interviews as admission process: A study on perception of health science students in Universiti Brunei Darussalam

Faiza Alam *[ID], Fatimah Az-Zahraa[º], Fazean Idris[º], Hanif Abdul Rahman[º]

Pengiran Anak Puteri Rashidah Sa'adatul Bolkiah Institute of Health Sciences, Universiti Brunei Darussalam, Bandar Seri Begawan, Brunei

º These authors contributed equally to this work.
* faiza.orakzai@gmail.com

**Data Availability Statement:** The data supporting the findings of this study is not publicly available due to restriction in institutional policy, however, they are available upon reasonable request from

## Abstract

### Background

As multiple mini-interviews (MMI) have grown in popularity in selecting applicants to health science programmes, it is essential to comprehend students' views towards MMI and its usefulness in the admissions process. The study aims to explore students' perceptions on the usefulness and satisfaction of the MMI as an admission process into the health science programmes offered in Universiti Brunei Darussalam (UBD).

### Methods

A cross-sectional study using a self-administered questionnaire was distributed to all Year 1 and 2 health sciences students in the university. For qualitative data, the responses obtained from the answers to the open-ended questions were analyzed manually using thematic analysis. Open coding was performed initially to identify words with similar meanings, recurring patterns and ideas. Focus coding was performed to group codes which sounded and felt similar. Themes were formulated and then reviewed. Descriptive statistics and univariate analysis were applied.

### Results

127 students participated in this study (53.4% response rate). 38.6% underwent the online MMI and 61.4% had the face-to-face MMI. 81% of participants agreed that the MMI was a fair assessment tool (81%), with adequate time allocated (91%), provided opportunity to demonstrate understanding of the profession (81%), and to express themselves (81%) but only 27% agreed that MMI was an enjoyable experience. No statistically significant differences were detected by gender. Those who did online MMI (89%) reported higher opportunity to express themselves, where 80% agreed it was an effective tool to assess selection of students in the health science profession. Univariate analysis revealed that male

the institutional ethics committee [rec.ihs@ubd. edu.bn].

**Funding:** The author(s) received no specific funding for this work.

**Competing interests:** All researchers, Dr Faiza Alam, Fatimah Az-Zahraa, Dr Fazean Irdayati Idris, and Dr Hanif Abdul Rahman would declare their conflict of interest with the participants as all four mentioned are either studying with or teaching the participants. This will not affect the study results by any means. This does not alter our adherence to PLOS ONE policies on sharing data and materials.

participants (42%) and those who did the online MMI (40%) were slightly more satisfied than those who did it onsite.

## Conclusion

Student perception is important for MMIs because it provides valuable insights into the effectiveness and fairness of the interview process. Study participants' perceptions of the MMI were positive, with objective reports on its fairness, timing, and feasibility but felt unprepared for the interview process and less enjoyable. Online MMI is favoured more by participants who rated it higher effectiveness and adequate timing with a better understanding of stations.

## Introduction

Multiple Mini Interviews (MMI) are structured interview formats commonly used in admission processes, particularly in fields such as medicine and healthcare, to assess applicants' non-cognitive qualities such as communication skills, ethical decision-making, and professionalism. They have been widely accepted and carried out all over the world as an undergraduate admissions process for students within the health sciences programmes. These interviews for the selection of students have grown in importance as the study of health science is frequently viewed as a challenging and demanding course, with limited number of enrolment spaces. MMI was developed and modeled after the Objective Structured Clinical Examination [1] which comprises about 5–10 stations assessing each candidate's performance based on their cognitive and non-cognitive skills [2]. Essentially, each station has a single interviewer and lasts about 8–10 minutes each. Scenarios including moral dilemmas, communication skills, teamwork, the strength of discussion, or some other non-cognitive attributes are given to each station to assess each candidate's response to the varying scenarios [3]. One interviewer stays in the designated station throughout the MMI duration, which reduces the possibility of a halo effect. Across many universities, MMI has been extensively used because of its flexibility, acceptability, and its reliability in assessing different areas and candidates' qualities across different educational settings.

Studies on MMI have looked into its validity and other facets of the candidates' acceptance. In a study by Dowell et al., researchers developed a questionnaire of attitudinal statements that perceived fairness, validity, the difficulty of the entire MMI process and whether they preferred traditional interview or MMI, and it was found that the candidates preferred MMI [4]. Another study by Eva et. al also reiterated that candidates viewed the MMI as a fair and acceptable tool for admission to medical school [5]. The paper provides insight regarding the attributes that are significant for attainment in clinical training as MMI approach is a validated for selecting medical candidates, due to its ability to the prediction of validity for clinical skills and ethics-based measures. This study also emphasizes the significance of cognitive and non-cognitive qualities assessment in medical trainees and provides substantiation for the permanence of candidate performance during the MMI process [5].

Bergelson et al, explain in their study that MMIs create a fair and standardized assessment process by providing each applicant with an equal opportunity to demonstrate their abilities. Student perception feedback helps identify any potential biases, inconsistencies, or flaws in the interview process [6].

Literature suggests that by gauging student perception, admissions committees can evaluate whether the MMIs are effectively assessing the desired non-cognitive qualities. If students feel that the interview format accurately reflects the skills and attributes relevant to the profession, it strengthens the construct validity of the interviews [7]. Furthermore, student perception feedback is crucial for evaluating the effectiveness of the MMI process and making necessary improvements [7]. It helps admissions committees gather information on various aspects, such as the clarity of instructions, the quality of interview stations, the overall organization, and the relevance of the questions. Identifying areas that need refinement can lead to continuous improvement of the interview process [8]. Nonetheless, the interview process itself can also significantly impact the experience on how candidates perceive the program or institution. If applicants have a positive perception of the MMIs, it enhances their overall impression of the admission process and the institution itself. Conversely, negative perceptions may deter qualified candidates from accepting offers or pursuing further studies [9].

As MMI has grown in popularity, it is essential to comprehend students' views towards MMI and its usefulness in the admissions process. Student perception is important for MMIs because it provides valuable insights into the effectiveness and fairness of the interview process. Thus, in our study, we aimed to assess the depth and scope of the MMI process within the health sciences program at the largest university in the Brunei Darussalam, focusing on health science students' perceptions and understanding of the MMI as a component of the student admissions procedure. Furthermore, to evaluate the perceived utility and effectiveness of the MMI from the perspective of health science students, exploring their opinions on the relevance and impact of the MMI in the admissions process for the health sciences program at the aforementioned university.

## Methods

### Study design and duration

This is a cross-sectional study that utilized both qualitative and quantitative approaches. Mixed-methods strategy was chosen because it was considered most suitable for answering the study's research questions and achieving its objectives. The study was conducted over a one-month period from 26 January 2023 to 28 February 2023. This duration was chosen based on the time where the majority of the targeted health sciences students were available during the university period.

### Study population

All 238 students of Year 1 and Year 2 students who were enrolled in the Bachelor of Health Science (BHSc) undergraduate programmes (i.e. Medicine, Dentistry, Biomedical Science, Pharmacy and Nursing/Midwifery) in PAPRSB Institute of Health Sciences (PAPRSB IHS), UBD from 2021 and 2022 were included in this study. After the ethics approval, an online consent was obtained from all participants. A link was provided to the participants to obtain online consent prior to completing the survey. Considering the small number of students in the institute, all Year 1 and Year 2 health science students will be recruited without sampling. Once the potential participants signed the consent form, they were redirected to the study questionnaire through the same link. In this study, only the health sciences students from PAPRSB IHS UBD were included, while data from other health sciences students from another institute such as Politeknik Brunei were excluded. Subjects who refused to give consent or are below the age of 18 years were also excluded.

## Sample size

A sample size of at least 132 is required to achieve a precision (power) of 5% (d = 0.05) on a population size of approximately 200 health science students in Year 1 and 2 of all disciplines in PAPRSB IHS, with the expected proportion of 50% at 95% confidence level. Accounting for attrition and missing data, a minimum of 132 responses are expected to fill out the questionnaire [10]. In this study, only 127 participants completed the questionnaire. The required sample size is/was 404 for the margin of error or absolute precision of ±5% in estimating the prevalence with 95% confidence. With this sample size, the anticipated 95% CI is/was (65%, 75%). This sample size is calculated using the Scalex SP calculator [10]. The number decreased down from 404 to 238 because of the terminated and drop-outs of the programme.

## Research instrument

The study described in this paper utilized a questionnaire with sections covering fundamental demographic data as well as a quantitative and qualitative assessment of participants' impression of MMI. The questionnaire comprises of 26 items; 7 questions about the participants demographics; 17 questions about how participants perceived their MMI experiences, and 2 questions for general feedback and suggestion.

Seven questions in the first component of the survey recorded socio-demographic information, such as gender, age, academic year, programme in IHS, and educational background. The second section (closed questions) included 13 questions that asked the participants to rate different facets of their MMI experience based on validity, reliability, fairness and content of MMI. Participants rated thirteen items using a 5-point Likert scale of 1 (strongly agree) to 5 (strongly disagree) and rated one item using a 3-point Likert scale of 1 (not satisfied) to 3 (satisfied). The third section comprised five open-ended questions in the final portion that invite participants to include further comments on their overall experience, their perceptions in general and suggestions for improvement.

This instrument had been validated in an earlier study whereby the closed-ended questionnaire was adapted from the survey used in a study conducted by McAndrew and Ellis [11]. Meanwhile the open-ended questionnaire that was utilized to elicit students' opinions about the MMI was adapted from previously published research [12]. The published questionnaires were modified from the published ones to accommodate for health science students enrolled in PAPRSB IHS UBD.

## Data collection

All data were collected anonymously via an online questionnaire using Qualtrics®. All statistical analyses were performed in the Macintosh® environment. Analysis was performed with Microsoft Excel® for statistical calculations.

## Statistical analyses

For quantitative data, the statistical analysis included descriptive statistics to describe participants' socio-demographic characteristics, participants' perception, and level of use of MMI during selection in the admission process based on the level of agreement of the scales. Subgroup analysis using Chi-square test for independence was used to investigate the association between study outcomes (participants' perception and level of satisfaction) by gender and mode of interview. All statistical analysis was computed using R v4.3.3®. A p-value less than 0.05 is considered statistically significant.

For qualitative data, the responses obtained from the answers to the open-ended questions were analyzed manually using thematic analysis. Open coding was performed initially to identify words with similar meanings, recurring patterns and ideas. Focus coding was performed to group codes which sounded and felt similar. Themes were formulated and then reviewed.

### Ethical considerations

Ethical approval to conduct the study was provided by the Institute of Health Science Research Ethics Committee and University Research Ethics Committee (IHSREC) UBD in December 2022 prior to data collection from participants. (Reference: UBD/PAPRSBIHSREC/2022/111). The study was carried out in accordance with the relevant guidelines and regulations as stipulated by the IHSREC of Universiti Brunei Darussalam.

## Results

### Quantitative data analysis

A total of 127 participants (53.4% response rate) completed the questionnaire. **Table 1** presents the demographic characteristics of the participants. They were mostly female students (77.2%) with an average age of 20 (±0.95) years old. 60.6% were second year undergraduates while the remaining were in Year 1. The participants included students enrolled in Medicine (40.2%), Nursing and Midwifery (32.3%), Pharmacy (11.0%), Biomedical Science (9.5%), and Dentistry

**Table 1. Participants' demographic data (n = 127).**

|  | n (%) |
|---|---|
| Mean Age (Years) [Mean (SD)] | 20 (0.95) |
| Participants' gender |  |
| Male | 29 (22.8%) |
| Female | 98 (77.2%) |
| Participants' academic year |  |
| Year 1 | 50 (39.4%) |
| Year 2 | 77 (60.6%) |
| Participants' programme |  |
| Pharmacy | 14 (11.0%) |
| Dentistry | 9 (7.1%) |
| Medicine | 51 (40.2%) |
| Nursing/Midwifery | 41 (32.3%) |
| Biomedical Science | 12 (9.5%) |
| Participants' educational background |  |
| International General Certificate of Secondary Education (IGCSE) | 6 (4.0%) |
| General Certificate of Education Advanced Level (GCE A'Level') | 115 (90.5%) |
| International Baccalaureate (IB) | 7 (5.5%) |
| Participants' who had previously attended a conventional interview |  |
| Yes | 99 (77.9%) |
| No | 26 (20.5%) |
| Would rather not say | 2 (1.6%) |
| Participants' mode of MMI |  |
| Online | 49 (38.6%) |
| Offline (F2F) | 78 (61.4%) |

**Table 2. Response to level of MMI usefulness by gender (Agree only).**

| Questionnaire item | Overall (n = 127) | Female (n = 98) | Male (n = 29) | P-value |
|---|---|---|---|---|
| Do you think it is important to be able to prepare for the stations? | 33 (28%) | 25 (28%) | 8 (29%) | 0.575 |
| Did you feel the MMIs gave you the opportunity to demonstrate your understanding of the profession? | 96 (81%) | 71 (79%) | 25 (89%) | 0.479 |
| Do you think the MMI is a fair assessment tool? | 95 (81%) | 71 (79%) | 24 (86%) | 0.569 |
| Do you think MMIs is an effective tool in assisting the selection of students in the health science profession? | 84 (71%) | 62 (69%) | 22 (79%) | 0.088 |
| For stations that you could prepare for, generally, did you performed well? | 81 (69%) | 60 (67%) | 21 (75%) | 0.072 |
| Did you feel the MMIs gave you the opportunity to express yourself? | 96 (81%) | 72 (80%) | 24 (86%) | 0.540 |
| Did you feel the MMIs gave you the opportunity to demonstrate your personal qualities? | 65 (55%) | 52 (58%) | 13 (46%) | >0.999 |
| Do you think the total length of time given for one interview session is enough? | 81 (69%) | 63 (70%) | 18 (64%) | 0.765 |
| Do you think the total length of time given for the whole MMI process is enough? | 52 (44%) | 43 (48%) | 9 (32%) | 0.465 |
| Do you think the total length of time given to read the questions and prepare for the station is enough? | 107 (91%) | 82 (91%) | 25 (89%) | 0.728 |
| Was the information provided by the University sufficient to allow you to feel comfortable about the interview process as a whole? | 91 (77%) | 71 (79%) | 20 (71%) | 0.583 |
| Do you think the MMI process was enjoyable for you? | 32 (27%) | 22 (24%) | 10 (36%) | 0.714 |
| For stations that you could not prepare for, generally, did you performed well? | 43 (36%) | 37 (41%) | 6 (21%) | 0.353 |

Chi-square test for independence

(7.1%) programme. Approximately a third of all participants had online MMI (38.6%) and 78 (61.4%) of participants had Face-to-Face (F2F) or offline MMI session.

Table 2 shows the respondents who agreed with the level of MMI usefulness items by gender. Majority of the participants agreed that the MMI was a fair assessment tool (81%), time allocated for the session was adequate (91%), provided opportunity to demonstrate understanding of the profession (81%), and to express themselves (81%). Nevertheless, only 27% agreed that MMI was an enjoyable experience. Interestingly, only 28% felt it was important to prepare for the MMI stations. For those stations they did not prepare for, 36% reported that they did not perform well. No statistical significant differences were detected by gender.

Table 3 shows the level of usefulness of MMI by mode of interview. Participants who did the F2F MMI (92%) reported slightly higher proportion of adequacy time preparation for the station than those online (89%). However, participants who did online MMI (89%) reported higher opportunity to express themselves than F2F participants (77%). Additionally, those who did the online MMI felt it was more enjoyable (31%), felt they performed well in stations they did not prepare for (49%), and reported that MMI was an effective tool to assess selection of students in the health science profession (80%). Although participants who did the online MMI reported slightly higher proportion for almost all of the items, they were not statistically significant.

Table 4 shows the level of satisfaction with MMI by gender and mode of interview. Univariate analysis revealed that male participants (42%) and those who did the online MMI (40%) were slightly more satisfied with the session than those who did the F2F MMI.

## Qualitative data analysis

More than half of the participants (n = 88) provided responses to the open-ended questions. Firstly, the data was manually analyzed using thematic analysis to identify key themes and patterns in the data [13]. Any contradictions or inconsistencies in the data were identified, but these were resolved through data reduction by abstracting and simplifying the data. Overall, a few themes emerged from the data, which were grouped into categories. The participants' perception of the strength of the MMI process is divided into 4 categories namely fairness,

**Table 3. Response to level of MMI usefulness by mode of interview (Agree only).**

| Questionnaire items | Offline (n = 78) | Online (n = 49) | P-value |
|---|---|---|---|
| Do you think it is important to be able to prepare for the stations? | 23 (32%) | 10 (22%) | 0.171 |
| Did you feel the MMIs gave you the opportunity to demonstrate your understanding of the profession? | 58 (79%) | 38 (84%) | 0.769 |
| Do you think the MMI is a fair assessment tool? | 58 (79%) | 37 (82%) | 0.888 |
| Do you think MMIs is an effective tool in assisting the selection of students in the health science profession? | 48 (66%) | 36 (80%) | 0.035 |
| For stations that you could prepare for, generally, did you performed well? | 47 (64%) | 34 (76%) | 0.111 |
| Did you feel the MMIs gave you the opportunity to express yourself? | 56 (77%) | 40 (89%) | 0.772 |
| Did you feel the MMIs gave you the opportunity to demonstrate your personal qualities? | 39 (53%) | 26 (58%) | 0.679 |
| Do you think the total length of time given for one interview session is enough? | 49 (67%) | 32 (71%) | 0.068 |
| Do you think the total length of time given for the whole MMI process is enough? | 33 (45%) | 19 (42%) | 0.667 |
| Do you think the total length of time given to read the questions and prepare for the station is enough? | 67 (92%) | 40 (89%) | 0.145 |
| Was the information provided by the University sufficient to allow you to feel comfortable about the interview process as a whole? | 55 (75%) | 36 (80%) | 0.938 |
| Do you think the MMI process was enjoyable for you? | 18 (25%) | 14 (31%) | 0.835 |
| For stations that you could not prepare for, generally, did you performed well? | 21 (29%) | 22 (49%) | 0.688 |

Chi-square test for independence

assessment of non-cognitive, basic knowledge and candidates' qualities. While the perception of participants on weaknesses of the MMI process is divided into 5 categories: limited time, the potential for bias and stress, the station placement and tailored answers from candidates. These categories included and were supported by quotes and examples from the data. These findings regarding participants' perceptions are represented as a thematic framework in Table 5 with illustrative quotes.

Based on the participants' responses provided when asked "*If you were unsatisfied, please comment on why*", it is evident that they had mixed feelings about their performance. While some admitted to not doing well, others felt that they could have done better had they been given more information about the test. In addition, some participants found the interview process to be overwhelming and intimidating, particularly when it was conducted online, with frequent station changes.

**Table 4. Participants' level of satisfaction with MMI.**

| | Satisfied (%) | P-value |
|---|---|---|
| **Overall** | 38% | - |
| **Gender** | | |
| Female | 36% | 0.169 |
| Male | 42% | |
| **Mode of interview** | | 0.596 |
| Offline | 37% | |
| Online | 40% | |

Chi-square test for independence

**Table 5. Participants' perception of MMI process.**

| | Sample Quotes |
|---|---|
| **Strength of MMI process** | |
| Fairness | *The outcome of the interview is based on multiple interviewers rather than just one. It gives more fairness all around.*<br>*Fair assessment as there were different interviewers instead of one*<br>*Able to explore into the student's critical thinking, their passion towards their chosen their programme, their personality based on how they answer and their knowledge* |
| Assessment of non-cognitive skills | *I think it's a good tool to use because it test the students knowledge and logical thinking under pressure, which is absolutely a great trait to have that is mostly use in healthcare profession*<br>*I believe communication skills and how we tackle anxiety ridden situations are important.*<br>*it can test the participants ability to communicate well and present their ideas to the interviewer* |
| Assessment of basic knowledge | *It encourages (forces) you to do prior research which means you'll end up accumulating more knowledge on your field of interest*<br>*I think asking questions regarding Brunei healthcare was one of the best tools as it was a good way to assess students whether they have done their research regarding it or not. If they did, it shows that they are enthusiastic on pursuing medicine.*<br>*It greatly demonstrates the participants determination in actually being a part of the course.* |
| Assessment of candidates' qualities | *this will help the interviewer see if the participants are actually capable of the programme based on their answers and thinking*<br>*To see who is more qualified or fit for the programme*<br>*Helps to narrow the number of students entering the programme if majority of the students have the required grades. It also helps to identify students who are actually interested and have some knowledge on the programme. It gives a chance to the students who don't get the grades but are able to talk.*<br>*It allows student to express their personal qualities other than academic credentials* |
| **Weaknesses of MMI process** | |
| Limited time | *Candidates may feel tired as they go on and on to the next interview with a new question/scenario given, which may affect their performance in the MMI.*<br>*Limited time to assess their depth of knowledge on certain topics as it is very fast-paced.*<br>*there is not enough time between the stations, we need time to calm down and then then time to read, then time to prepare.*<br>*Limited time available to think of the answer before discussing it with the interviewer* |
| Potential of bias | *there could be potential of bias, the interviewer may have some sort of prejudice, which can affect the assessment of candidates* |
| Potential for stress | *its a stressful situation for students especially when you cant stop thinking about what you said in the previous station that was absolutely wrong, and this affects the station after as the student become distracted with the thought of messing up*<br>*The pressure you get just being in the presence of other participants from different backgrounds. Especially in nursing, you are in competition with participants who were previously in diploma program.*<br>*a long process, the interviewee may get demotivated especially after a badly-done station and still need to continue with other stations afterwards. they might still haven't got time to recover from the previous station* |
| Station placement | *The student in a resting station can listen to whatever is being said/presented in the neighboring station, which can interrupt thought process.*<br>*Sometimes it can be confusing as students need to move from one room to another. Even though it is quite straightforward and there is also a person to guide you where to go next, during an interview when you're nervous, it makes you even more panicky and it does not give you some time to calm down your thoughts in between sessions.* |
| Tailored answers | *Some questions could've been prepared for earlier then memorized answer rather than giving a critically thought answer*<br>*Interview questions are redundant and predictable. Candidates are good in "marketing" themselves but in reality, they are not competent for hands on skills.*<br>*but at the same time students might exaggerate their answer to be seen as one of the best qualified candidates* |

One of the participants expressed the need for more scenario-based questions. Such feedback is useful to the programme, as inclusion of such questions can help assess one's thought processes, particularly their ability to think on their feet and respond to unexpected situations. Furthermore, this can potentially assist interviewers to determine whether or not the candidate has the necessary qualities to become a good doctor, such as empathy, compassion, and a genuine concern for the well-being of their patients.

## Strength of the MMI process

According to the participants' responses, the MMI process's strengths lay in its fairness and non-bias as it allows multiple interviewers to evaluate the candidate, as opposed to just one. In addition, it can assess the candidate's non-cognitive skills, including communication, critical thinking, decision-making, and performance under stress. It also allows for the assessment of the candidate's personal qualities and abilities, demonstrating their knowledge and understanding of the program and profession. Furthermore, the process helps the interviewer to recognize which students will perform well in the programme in the future based on their overall performance.

A further advantage of the MMI process is that it encourages prior research and outside reading, allowing the candidate to acquire more knowledge in their area of interest. This helps in assessing the candidate's knowledge on a broader scope than just grades alone. In addition, the university is able to assess the student's character and demeanor under specific circumstances, which can indicate the type of healthcare professional the student will become in the future and observe their determination and passion towards pursuing the profession.

## Weaknesses of the MMI process

Although MMI has been widely adopted as one of the tools for selecting students for health science programmes, it does have some limitations. One of the most frequent complaints is that candidates are given insufficient time to rest, which can result in fatigue and a decline in performance at subsequent stations. The rapid pace of the MMI has also been identified as a significant flaw. Some participants believe they lack sufficient time to adequately prepare for the interviews, which may result in apprehension and subpar performance. Additionally, the time allotted for reading questions is often too short, which can lead to students feeling rushed and stressed.

Another shortcoming of MMI lies in its potential to be biased of MMI. It is possible that interviewers have biases that influence their evaluations of candidates, which could result in unjust outcomes. On top of that, candidates who are socially awkward or shy may be at a disadvantage during the MMI process, as they may struggle to express themselves clearly and confidently. The MMI process can also be stressful for participants, and this could affect their disposition and self-assurance throughout the interviews.

Besides that, the limited time available to access candidates is a significant disadvantage of the MMI procedure. There may not be enough time for interviewers to thoroughly evaluate the non-cognitive qualities that are essential in the health science profession. Similarly, students may not be able to fully demonstrate their personality, identity, and skills during the interviews due to time constraints. In some cases, candidates may also have tailored answers, which may not reflect their true personalities or knowledge.

## Participants' feedback

The responses from participants provided several suggestions for improving the MMI process for potential candidates in the future. One common suggestion was to reduce the number of

stations to make the process less overwhelming and more manageable for candidates. Additionally, some students proposed the addition of general knowledge questions pertaining to current events and challenges to access critical thinking skills. While other students recommended scenario-based questions and character-based questions to evaluate if the candidate is suitable for the profession. Apart from that, they also suggested that the university should provide more information about the interview process, including an overview of the topics and questions that will be covered, so that students can prepare adequately. Furthermore, allowing more time for reading and preparing for each station is suggested, as well as rest and recuperation time between stations.

Another suggestion was to create a more relaxed and friendly environment for the interview, with more conversational-style questions and practical skills assessment. In order to provide a more private and less distracting environment, they also recommended having different rooms for each station rather than a large room with curtains as partition.

Finally, some participants suggested that the interviewers should be more attentive, provide more time for candidates to answer questions, and avoid being condescending or discouraging during the interview. They recommended that interviewers give constructive feedback and encourage candidates to express their opinions and showcase their skills. Overall, the suggestions provided valuable insights into improving the MMI process to make it more efficient and less stressful for prospective candidates.

## Participants' personal experience with MMI

It appears that the experience of MMI can vary greatly depending on several factors. The format of the interview, whether it is conducted online or F2F, can influence the technical difficulties that may arise and the overall experience. Due to possible technical difficulties, some participants may find the online format more difficult, whereas others may prefer the F2F experience. Interview preparation can also play a role in the experience. Participants who feel unprepared for certain questions may experience a lack of confidence, while others who feel well-prepared may feel more assured. The quality of the interviewer can also influence the experience, as a more engaging interviewer may help to alleviate nervousness and encourage individuals to respond to queries with greater confidence.

The MMI experience can elicit a range of emotions, from positive and enjoyable to negative and traumatic. Participants shared that the presence of rest stations can help them, the candidates to mentally prepare and ease nervousness.

Overall, majority of participants shared that the MMI had been a unique and challenging experience that can vary significantly depending on several factors. Nevertheless, it provides insight into the attitudes and aspects that are valued in selected professions, and preparations to help them advance and perform well in future interviews.

## Discussion

This study investigated the perceptions and understanding of the Multiple Mini Interview (MMI) and its utility among health science students in UBD using a questionnaire-based format, while qualitatively gathering their feedback on the strengths, weaknesses and suggestions for improvements of the MMI.

Based on participants' responses from the questionnaires, it was interesting to note that only 28% of participants felt it was important to prepare for the MMI stations, while only 27% agreed that MMI was an enjoyable experience. In other studies investigating MMI as an effective tool for admissions, participants reflected that MMI was an innovative idea for a non-biased selection process and that their experience of it was enjoyable, relaxing and positive

[14] while providing participants with better opportunities to portray themselves [15]. While another study reiterated that the MMI was a fair assessment process compared to the traditional interview and that non-cognitive skills were evaluated more effectively through MMI [16] This was in-keeping with participants' qualitative responses on the MMI's strengths and weaknesses, where some participants agreed that MMI facilitated a more objective evaluation of candidates due to various evaluators assessing different aspects of a candidate's performance. They also noted that as each candidate was evaluated by the same group of evaluators, suggesting that the MMI process allowed for a fair and more objective evaluation of all candidates. Meanwhile, in another study by Hopson et al, participants were in favor of a combination of MMI and unstructured interaction when compared to MMI alone, where a lower MMI performance was associated with higher preference of the traditional interview while a higher MMI performance was associated with higher MMI preferences [17].

A majority of our study participants concurred that the total time allocated for the MMI process was sufficient. Similar to our study, Hofmeister et. al found that their candidates felt they had sufficient time for the MMI, requiring no specialized knowledge to express themselves fully within the stations [18]. However, this was in contrast to their responses to the open-ended questions when asked about the weaknesses of MMI, where participants mentioned that the time allotted for reading the questions and preparing for the next station is frequently insufficient, resulting in anxiety and subpar performance as they felt rushed and stressed. The study by Kumar et. al also reported that participants felt stressed as the time was short and that they had to change their thought process after every other station [12]. MMIs typically involve multiple stations or scenarios, each with a limited time frame for the applicant to respond. This time pressure can increase stress levels and make the process feel intense. This was also observed in the study conducted by Corelli, where the students expressed stress related to MMI more than in standard interviews [19], suggesting that increasing the time allocated for preparing and perusing the questions could enhance the interviewee's perception and performance. In our study, candidates were satisfied with the time allocated, thus suggesting that the scenarios for each station were adequately furnished for the interview duration.

The MMI format can indeed be perceived as stressful and intense by applicants. The uncertainty of what to expect in each station can contribute to heightened stress levels. The outcome of the MMI process is often crucial for applicants' admission into highly competitive programs, such as medical schools. The awareness of the high stakes involved can lead to increased pressure and intensify the perception of stress. Another reason for the stressful situation is unfamiliarity with the stations. In an MMI, applicants are usually assessed by multiple evaluators or interviewers at different stations. Having to interact with different individuals in rapid succession can add to the intensity of the experience, especially if applicants feel they are being constantly evaluated. MMIs often present applicants with various scenarios or ethical dilemmas that require quick thinking and decision-making. In the study conducted by Eva et al, it was expressed that having different interviewers at different stations might help recover from unsatisfactory performance in the previous station [20]. In our study, each station had one interviewer who remained throughout the duration of the MMI thus increasing the reliability of the assessment of that station and we used a total of five stations covering different themes, for a total of 60 candidates for that day. Meanwhile a study in Hamburg concluded the benefits of increasing the number of stations instead of raters within stations to ensure reliability, and that investing in scoring, rater training and scenario development would be cost-effective [21].

When comparing the online versus F2F mode of MMI, our study saw that more participants felt that the online MMI was a fair and effective assessment, with more adequate timing for the interview sessions. The online mode also provided better opportunities to demonstrate

their understanding and express their personal qualities, thus leading them to perform better in stations compared to the F2F MMI. Other studies have reported that internet-based MMI are well-received by candidates who rated them with high acceptability and satisfaction, with no significant differences in scores equivalence between conventional (onsite) MMI and internet-based MMI (online) [22]. Online-based methods are also cost-friendly and considered an important innovation that can be utilized by other universities [22]. A recent study done in a medical school in Indonesia supported that online MMI was found to be more feasible and cost-efficient [23].

In addition, our study revealed that participants preferred scenario-based questions for evaluating their critical thinking and reasoning skills, particularly in responding to unexpected situations. This may be in view of the fact that scenario-based questions can provide valuable insight into a candidate's empathy, compassion, and communication abilities. This finding is consistent with previous research suggesting that scenario-based inquiries can provide a more accurate assessment of candidates' abilities [5]. In addition, participants wished for additional information from the university in order to feel secure with the entire MMI procedure. Addressing these weaknesses can guarantee a fair and effective selection procedure that identifies the most qualified candidates for various health programmes.

Participants' expectations and understandings of the MMI's purpose varied according to the findings of the study. Some participants believed that the MMI is primarily used to evaluate candidates' non-cognitive skills, such as communication, as opposed to those who believed that the MMI is only used to evaluate candidates' knowledge of the profession and programme. When designing and implementing the MMI process in the future, it will be crucial to consider the student's perspective, as revealed by these findings. Students involved in the study conducted by Kelly et. al expressed that MMI gives a very clear insight at the time of admission of what is expected from them during medical school [24].

Overall, the perceptions of our participants in the MMI were positive, with objective reports on its fairness, timing, and feasibility. However, the interview process can nonetheless be daunting for any candidate, subjecting them to feeling that it was far less enjoyable and unprepared for the interviews, despite sufficient information provided by the university. Online MMI seemed to be favored more by participants who underwent this mode of interview and rated it higher in terms of fairness, effectiveness, adequate timing with a better understanding of stations.

## Limitations

There were limitations to this study, one of which was a small sample size that limits the generalizability of the findings. The response rate for this study is low, whereby only 127 out of 238 students participated in this study. In addition, the research was only conducted at a single university and may not be representative of other universities or health science programmes. In order to increase the generalizability of the findings, future studies could involve a larger sample size and multiple universities. Furthermore, despite excluding Year 3 and Year 4 to avoid recall bias, recall bias cannot be ensured from the included students. With the passage of time, the students may have changed their perceptions due to the learning environment provided at IHS from the actual perception from the time of their MMI. The time delay from the day of their MMI and the day they answered the questionnaire may have had an impact on how respondents rated their actual experience and perception.

## Conclusion

The purpose of this study was to obtain insight into how students perceive the MMI and its usefulness in student admission. Being the first study documenting the experiences and

comprehension of MMI among PAPRSB IHS UBD students, this study allows us to identify the strengths and weaknesses of MMI from the perspective of the student. The students consider the MMI process as a fair and efficient instrument for selecting candidates for the health science profession. These findings will be beneficial to the institution because they provide valuable insight into students' perceptions that will aid in making the process more student-friendly and ensuring that it accurately evaluates the skills and qualities necessary for success in their chosen field, thereby improving the quality of their students. It is important to note that student perception is just one component of the evaluation process, and decisions should not solely rely on it. However, considering student feedback and perceptions can contribute to a more comprehensive and reflective evaluation of the MMI process, leading to ongoing improvements and fairer admissions practices.

## Recommendations

The findings from this study have significant benefits for the institution, as they can inform the admissions process enhancement and ultimately contribute to the calibre of admitted students. By investigating the students' perceptions of the MMI, we can gain a better understanding of how to make the admission process more student-friendly and efficient. Furthermore, this profound selection can pick the keen students and separate them from the ones who barely have an interest towards a particular field.

## Acknowledgments

All authors would like to acknowledge all the participants of the study. They all participated after informed written consent.

## Author Contributions

**Conceptualization:** Faiza Alam.

**Data curation:** Fatimah Az-Zahraa, Fazean Idris.

**Formal analysis:** Hanif Abdul Rahman.

**Investigation:** Faiza Alam, Fatimah Az-Zahraa, Fazean Idris.

**Methodology:** Faiza Alam, Fazean Idris.

**Project administration:** Faiza Alam.

**Supervision:** Faiza Alam.

**Validation:** Faiza Alam, Fazean Idris.

**Writing – original draft:** Faiza Alam, Fatimah Az-Zahraa, Fazean Idris, Hanif Abdul Rahman.

**Writing – review & editing:** Faiza Alam, Fatimah Az-Zahraa, Fazean Idris, Hanif Abdul Rahman.

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
