## [Decision Letter · Decision Letter 0]

28 Nov 2023

PONE-D-23-20665Multiple mini-interviews as admission process: a study on perception of health science students in Universiti Brunei DarussalamPLOS ONE

Dear Dr. Alam,

Thank you for submitting your manuscript to PLOS ONE. After careful consideration, we feel that it has merit but does not fully meet PLOS ONE’s publication criteria as it currently stands. Therefore, we invite you to submit a revised version of the manuscript that addresses the points raised during the review process.

Please see the comments from one reviewer below. Please note that we have only been able to secure a single reviewer to assess your manuscript. We are issuing a decision on your manuscript at this point to prevent further delays in the evaluation of your manuscript. Please be aware that the editor who handles your revised manuscript might find it necessary to invite additional reviewers to assess this work once the revised manuscript is submitted. However, we will aim to proceed on the basis of this single review if possible. 

Please ensure that your work is thoroughly copyedited before resubmission. We note that there are grammatical errors in the title and throughout the manuscript, and that there is a variation of tenses used in the manuscript.

Please ensure that the generalisability of the findings beyond the case study used is clear - PLOS ONE does not publish case studies, and results must add to the existing body of academic knowledge.

We look forward to receiving your revised manuscript.

Kind regards,

Hanna Landenmark

Staff Editor

PLOS ONE

Journal Requirements:

"I have read the journal's policy and the authors of this manuscript have the following competing interests: [All researchers, Dr Faiza Alam, Dr Fazean Irdayati Idris, and Dr Hanif Abdul Rahman would declare their conflict of interest with the participants as all the four mentioned are either studying with or teaching the participants. This will not affect the study results by any means]."

Reviewers' comments:

Reviewer's Responses to Questions

**Comments to the Author**

1. Is the manuscript technically sound, and do the data support the conclusions?

Reviewer #1: Yes

2. Has the statistical analysis been performed appropriately and rigorously? 

Reviewer #1: Yes

3. Have the authors made all data underlying the findings in their manuscript fully available?

Reviewer #1: Yes

4. Is the manuscript presented in an intelligible fashion and written in standard English?

Reviewer #1: Yes

5. Review Comments to the Author

Reviewer #1: This is a useful work particularly relevant to new schools or those that are implementing the MMI for the first time. Institutions, where the MMI is well established, would find the work applicable to tracking the quality of stations as seen by the applicants, which is important information. The authors correctly point out that the experience during the admissions process can affect the reputation of the school and the decision of students to attend or select another school. The authors use a combination of qualitative and quantitative statistical approaches, which, as is noted, are necessary given the number of objectives.

Comment 1: The paper would benefit if the authors provided a list of objectives,

Although a complex study was well designed, it suffered from a poor response, which brought the numbers to a lower threshold. Nevertheless, it does show the strengths and weaknesses of the MMI.

The authors should provide more information in the study on “preparing for the MMI”. The following questions need clarification.

Did the applicants have the specific stations that they encountered in the MMI or were they given a generic set of questions?

How far in advance did they receive the stations?

Were they instructed on limitations in discussing their stations with fellow applicants or others?

Did all applicants have access to the same quality of libraries?

The authors noted that, depending on preparation for the MMI some applicants were more comfortable with stations than other participants in the study. Was the degree of comfort with a specific station a predictor of performance? Was the degree of comfort with the entire process a predictor of performance? Again, this would be useful information for planners of an MMI

6. PLOS authors have the option to publish the peer review history of their article (what does this mean?). If published, this will include your full peer review and any attached files.

Reviewer #1: No

---

## [Author Response · Author response to Decision Letter 0]

7 Jan 2024

Comments to the Author

Reviewer 1

Comment: 

The paper would benefit if the authors provided a list of objectives.

Response:

Objectives have been added at the end of the introduction section. (Page 6, Lines107 – 112)

Comment: 

Although a complex study was well designed, it suffered from a poor response, which brought the numbers to a lower threshold. Nevertheless, it does show the strengths and weaknesses of the MMI.

The authors should provide more information in the study on “preparing for the MMI”. The following questions need clarification.

Did the applicants have the specific stations that they encountered in the MMI or were they given a generic set of questions?

Response:

They were given a generic set of questions. In the MMI, students were tested in ten individual stations that assessed the following skills:

A: Students’ motivation, qualities and achievements, communication, and self-reflection

B: Knowledge of Health Care and Current Issues

C: Ethics

D: Critical Thinking

Comment:

How far in advance did they receive the stations?

Response:

They were briefed by giving a set of instructions and information on the themes of the stations via email 2 weeks before the MMI.

They were given 5 minutes to familiarise themselves with the stations before each station began.

Comment:

Were they instructed on limitations in discussing their stations with fellow applicants or others?

Response:

The applicants were instructed not to discuss the stations with their peers.

Comment:

Did all applicants have access to the same quality of libraries?

Response:

Applicants came from different prior institutions, and therefore quality of libraries would differ.

Applicants' preparations for the MMI were dependent on their ability to prepare from various sources and not restricted to any libraries. General recommendations for the MMI preparation were made without giving specific access to libraries.

Comment:

The authors noted that, depending on preparation for the MMI some applicants were more comfortable with stations than other participants in the study. Was the degree of comfort with a specific station a predictor of performance? 

Response:

Rather than comfort, MMI assessors provided a score for the global rating of the applicant during the interview. This score was based on the assessor's overall impression of the candidate, specifically looking at their communication skills and how articulated their responses were, how confident and composed they were, their professional appearance and demeanour, as well as motivation and enthusiasm during their time spent in each station, and scored separately by respective assessors pertinent to that station.

Comment:

Was the degree of comfort with the entire process a predictor of performance? Again, this would be useful information for planners of an MMI.

Response:

In our Institute's interview process, the global rating was not factored in the total scoring of applicants' performance, but rather as a deciding factor when evaluating borderline applicants (for example if two or more applicants had the same scores, the global rating will assist in making the final decision between the two).

The detailed process of MMI at PAPRSB Institute of Health Sciences UBD has been previously published separately. 

Citation:

Alam F, Lim YC, Chaw LL, Idris F, Kok KYY. Multiple mini-interviews is a predictor of students' academic achievements in early undergraduate medical years: a retrospective study. BMC Med Educ. 2023 Mar 27;23(1):187. doi: 10.1186/s12909-023-04183-7. PMID: 36973779; PMCID: PMC10044430.

---

## [Decision Letter · Decision Letter 1]

6 May 2024

PONE-D-23-20665R1Multiple mini-interviews as admission process: a study on perception of health science students in Universiti Brunei DarussalamPLOS ONE

Dear Dr. Alam,

Thank you for submitting your manuscript to PLOS ONE. After careful consideration, we feel that it has merit but does not fully meet PLOS ONE’s publication criteria as it currently stands. Therefore, we invite you to submit a revised version of the manuscript that addresses the points raised during the review process.

As observed in the referee evaluations of the article, providing more detailed explanations, especially regarding the analysis of qualitative data, is important for the reliability and validity of the research.
The use of quantitative and qualitative data together requires evaluation with mixed research methodology.
It would be appropriate to consider the discussion and conclusion sections within this framework.
I believe that it would be appropriate to revise and edit the article for these reasons.:Indicate which changes you require for acceptance versus which changes you recommendAddress any conflicts between the reviews so that it's clear which advice the authors should followProvide specific feedback from your evaluation of the manuscriptPlease ensure that your decision is justified on PLOS ONE’s publication criteria and not, for example, on novelty or perceived impact.

We look forward to receiving your revised manuscript.

Kind regards,

Ayse Hilal Bati, Professor

Academic Editor

PLOS ONE

Journal Requirements:

Additional Editor Comments:

Dear Author/s,

As observed in the referee evaluations of the article, providing more detailed explanations, especially regarding the analysis of qualitative data, is important for the reliability and validity of the research. The use of quantitative and qualitative data together requires evaluation with mixed research methodology. It would be appropriate to consider the discussion and conclusion sections within this framework. I believe that it would be appropriate to revise and edit the article for these reasons.

Reviewers' comments:

Reviewer's Responses to Questions

**Comments to the Author**

1. If the authors have adequately addressed your comments raised in a previous round of review and you feel that this manuscript is now acceptable for publication, you may indicate that here to bypass the “Comments to the Author” section, enter your conflict of interest statement in the “Confidential to Editor” section, and submit your "Accept" recommendation.

Reviewer #2: All comments have been addressed

Reviewer #3: (No Response)

2. Is the manuscript technically sound, and do the data support the conclusions?

Reviewer #2: Yes

Reviewer #3: No

3. Has the statistical analysis been performed appropriately and rigorously? 

Reviewer #2: Yes

Reviewer #3: No

4. Have the authors made all data underlying the findings in their manuscript fully available?

Reviewer #2: Yes

Reviewer #3: No

5. Is the manuscript presented in an intelligible fashion and written in standard English?

Reviewer #2: Yes

Reviewer #3: No

6. Review Comments to the Author

Reviewer #2: All the reviewer's comments have been addressed satisfactorily. There is good improvement in this revised paper.

Reviewer #3: Dear author,

The statistcial data analysis usedin this paper is wrong and not suitable for the data.

APA was not followed in the paper.

The discussion is poorly written and does not address the previous reserach.

7. PLOS authors have the option to publish the peer review history of their article (what does this mean?). If published, this will include your full peer review and any attached files.

Reviewer #2: No

Reviewer #3: **Yes: **Houman Bijani

---

## [Author Response · Author response to Decision Letter 1]

23 Jul 2024

Date: May 06 2024 12:57AM

Subject: PLOS ONE Decision: Revision required [PONE-D-23-20665R1]

EDITOR COMMENT 1: 

As observed in the referee evaluations of the article, providing more detailed explanations, especially regarding the analysis of qualitative data, is important for the reliability and validity of the research. The use of quantitative and qualitative data together requires evaluation with mixed research methodology. It would be appropriate to consider the discussion and conclusion sections within this framework. I believe that it would be appropriate to revise and edit the article for these reasons.

Response:

Thank you for your valuable input and we agree with the comment. This paper has utilised mixed methods approach, and has interpretation of the findings are that of complementary of each method, following through in the discussion and conclusion. Please find the revised manuscript submission. Thank you for your kind reconsideration and we hope to receive your favourable decision soon.

REVIEWER # 2 COMMENTS:

All the reviewer's comments have been addressed satisfactorily. There is good improvement in this revised paper.

Response:

Thank you for your valuable comment, we greatly appreciate it.

REVIEWER # 3 COMMENTS:

Comment #1:

The statistical data analysis used in this paper is wrong and not suitable for the data.

Response:

Thank you for your valuable comment. We would appreciate further clarification on this point. Currently, we are comparing a categorical factor towards a categorical outcome, and the chi-square test is appropriately used in this univariable inference. Due to the limitation of a small sample size, higher order analysis was not possible and this is acknowledged.

Comment #2:

APA was not followed in the paper.

Response:

PLOS One requirement is Vancouver and not APA style. Thus, APA has not been followed in the manuscript.

Comment #3:

The discussion is poorly written and does not address the previous research.

Response:

We have revised the discussion section as per the comment. 

Kindly find our revised manuscript submission.

---

## [Decision Letter · Decision Letter 2]

14 Oct 2024

PONE-D-23-20665R2Multiple mini-interviews as admission process: a study on perception of health science students in Universiti Brunei DarussalamPLOS ONE

Dear Dr. Alam,

Thank you for submitting your manuscript to PLOS ONE. After careful consideration, we feel that it has merit but does not fully meet PLOS ONE’s publication criteria as it currently stands. Therefore, we invite you to submit a revised version of the manuscript that addresses the points raised during the review process.

 In addition to the minor suggestion made by Reviewer #4 below, please also ensure that: - APA formatting has been fully followed throughout the paper 

- the discussion sufficiently incorporates previous supporting literature

We look forward to receiving your revised manuscript.

Kind regards,

Ayse Hilal Bati, Professor

Academic Editor

PLOS ONE

Journal Requirements:

Additional Editor Comments :

Thank you

Reviewers' comments:

Reviewer's Responses to Questions

**Comments to the Author**

1. If the authors have adequately addressed your comments raised in a previous round of review and you feel that this manuscript is now acceptable for publication, you may indicate that here to bypass the “Comments to the Author” section, enter your conflict of interest statement in the “Confidential to Editor” section, and submit your "Accept" recommendation.

Reviewer #2: (No Response)

Reviewer #3: (No Response)

Reviewer #4: All comments have been addressed

2. Is the manuscript technically sound, and do the data support the conclusions?

Reviewer #2: Yes

Reviewer #3: Partly

Reviewer #4: Yes

3. Has the statistical analysis been performed appropriately and rigorously? 

Reviewer #2: Yes

Reviewer #3: No

Reviewer #4: Yes

4. Have the authors made all data underlying the findings in their manuscript fully available?

Reviewer #2: Yes

Reviewer #3: Yes

Reviewer #4: Yes

5. Is the manuscript presented in an intelligible fashion and written in standard English?

Reviewer #2: Yes

Reviewer #3: No

Reviewer #4: Yes

6. Review Comments to the Author

Reviewer #2: All the reviewers' comments have been addressed satisfactorily. There is good improvement in this second revised paper.

Reviewer #3: (No Response)

Reviewer #4: Although this study is not the first in its field, it is well organized and completed in a way that includes elements which can provide ideas for the future. In addition, the authors have revised the text by taking into account the suggestions of previous reviewers. It is seen that these revisions are sufficient.

Only the table number given in lines 239-240 is incorrect. It is the table where the quotations are given and it is called Table 4, but it should be Table 5.

7. PLOS authors have the option to publish the peer review history of their article (what does this mean?). If published, this will include your full peer review and any attached files.

Reviewer #2: No

Reviewer #3: **Yes: **Houman Bijani

Reviewer #4: No

---

## [Author Response · Author response to Decision Letter 2]

20 Oct 2024

Date: Oct 14 2024 05:18AM

Subject: PLOS ONE Decision: Revision required [PONE-D-23-20665R2]

EDITOR COMMENT 1: 

In addition to the minor suggestion made by Reviewer #4 below, please also ensure that:

- APA formatting has been fully followed throughout the paper 

- the discussion sufficiently incorporates previous supporting literature.

Response:

APA formatting has been fully followed throughout the paper .

REVIEWER # 2 COMMENTS:

All the reviewer's comments have been addressed satisfactorily. There is good improvement in this revised paper.

Response:

Thank you for your valuable comment, we greatly appreciate it.

REVIEWER # 3 COMMENTS:

No Response 

REVIEWER # 4 COMMENTS:

Although this study is not the first in its field, it is well organized and completed in a way that includes elements which can provide ideas for the future. In addition, the authors have revised the text by taking into account the suggestions of previous reviewers. It is seen that these revisions are sufficient.

Only the table number given in lines 239-240 is incorrect. It is the table where the quotations are given and it is called Table 4, but it should be Table 5.

Response:

The table number has been edited.

---

## [Editor Report · Decision Letter 3]

18 Nov 2024

Multiple mini-interviews as admission process: a study on perception of health science students in Universiti Brunei Darussalam

PONE-D-23-20665R3

Dear Dr. Alam,

We’re pleased to inform you that your manuscript has been judged scientifically suitable for publication and will be formally accepted for publication once it meets all outstanding technical requirements.

Kind regards,

Ayse Hilal Bati, Professor

Academic Editor

PLOS ONE

Additional Editor Comments (optional):

Dear Authors,

Thanks for your research.
---

## [Editor Report · Acceptance letter]

22 Nov 2024

PONE-D-23-20665R3 

PLOS ONE

Dear Dr. Alam, 

I'm pleased to inform you that your manuscript has been deemed suitable for publication in PLOS ONE. Congratulations! Your manuscript is now being handed over to our production team.

Kind regards, 

on behalf of

Dr. Ayse Hilal Bati 

Academic Editor

PLOS ONE